# Endoplasmic Reticulum Stress in Chemotherapy-Induced Peripheral Neuropathy: Emerging Role of Phytochemicals

**DOI:** 10.3390/antiox11020265

**Published:** 2022-01-28

**Authors:** Yugal Goel, Raghda Fouda, Kalpna Gupta

**Affiliations:** 1Hematology/Oncology, Department of Medicine, University of California, Irvine, CA 92697, USA; ygoel@hs.uci.edu (Y.G.); foudar@hs.uci.edu (R.F.); 2VA Medical Center, Southern California Institute for Research and Education, Long Beach, CA 90822, USA; 3Department of Medicine, Division of Hematology, Oncology and Transplantation, University of Minnesota, Minneapolis, MN 55455, USA

**Keywords:** cancer, chemotherapy, endoplasmic reticulum stress, oxidative stress, neuropathy, pain, phytochemicals

## Abstract

Chemotherapy-induced peripheral neuropathy (CIPN) is a significant dose-limiting long-term sequela in cancer patients undergoing treatment, often leading to discontinuation of treatment. No established therapy exists to prevent and/or ameliorate CIPN. Reactive oxygen species (ROS) and mitochondrial dysregulation have been proposed to underlie the pathobiology of CIPN. However, interventions to prevent and treat CIPN are largely ineffective. Additional factors and mechanism-based targets need to be identified to develop novel strategies to target CIPN. The role of oxidative stress appears to be central, but the contribution of endoplasmic reticulum (ER) stress remains under-examined in the pathobiology of CIPN. This review describes the significance of ER stress and its contribution to CIPN, the protective role of herbal agents in countering ER stress in nervous system-associated disorders, and their possible repurposing for preventing CIPN.

## 1. Introduction

Cancer accounted for a million deaths and 19.3 million new cases in the year 2020 [1]. Chemotherapy is the mainstream treatment with/without surgery or radiotherapy, depending on the stage of the tumor [2]. However, it may cause prolonged and often irreversible debilitating chemotherapy-induced peripheral neuropathy (CIPN) in addition to nephropathy [3], cardiomyopathy [4], chemotherapy-induced lung disease [5], inhibition of the proliferative cells of the bone marrow [6], hair follicles [7], gut [8], urinary, and reproductive tracts [9,10]. Of these adverse effects of chemotherapy, CIPN is a significant dose-limiting long-term sequela associated with several commonly used chemotherapeutic agents as a first-line therapy against several solid and hematological malignancies [11]. CIPN remains one of the leading causes for dose reduction or discontinuation of chemotherapy during treatment [12]. It can start during the early phases of chemotherapy and may persist after discontinuation of therapy, impairing the quality of life for cancer survivors. The presentation may vary, including sensory, behavioral symptoms and alterations in gait. No established mechanism or therapy exists for CIPN, but oxidative stress appears to be central to the axonal injury instigated by chemotherapeutic agents, and several pharmacologic and nutraceuticals/phytochemicals are under investigation to prevent and/or ameliorate CIPN.

This review describes the existing knowledge and lacunae regarding the significance of ER stress and its contribution to CIPN, followed by a discussion on the protective role of phytochemicals in targeting ER stress with the potential to prevent/treat CIPN. We discuss ER stress in CIPN in the light of existing knowledge from other neuropathic conditions with some resemblance to CIPN and its targeting with promising phytochemicals under investigation. Additionally, we also provide a brief overview of the mechanism of action of chemotherapeutic agents, the symptoms of CIPN, and present treatment strategies to appreciate the critical role of ER stress in the context of CIPN.

## 2. Chemotherapy Induces Peripheral Neuropathy

Chemotherapeutic agents associated with the development of peripheral neuropathy consist of platinum-based compounds (cisplatin, carboplatin, and oxaliplatin) that inhibit DNA synthesis by inducing DNA cross-linking [13], and microtubule inhibitors (MTI) such as vinca alkaloids (Vincristine, Vinblastine, Vinorelbine, Vindesine, and Vinflunine), epothilones (ixabepilone) [14], and taxanes (Paclitaxel and Docetaxel) that act by interfering with microtubules, thereby blocking mitosis and cell cycle progression resulting in apoptosis of the cell. In addition to proteasome inhibitors (bortezomib) that induce cellular stress and apoptosis by perturbation of intracellular protein homeostasis [15], immunomodulatory drugs (Thalidomide, Pomalidomide, and Lenalidomide) perform antitumor activity by different mechanisms, including inhibition of the cytokines, tumor necrosis factor-α (TNF-α), T-cell costimulatory and antiangiogenic activities [11,16]. Because of the often-unavoidable widespread use of chemotherapy to treat cancer, CIPN remains a major challenge without established therapy to prevent or treat it [17]. Duloxetine has been recommended for the treatment of established CIPN but lacks the support of mechanistic evidence with limited beneficial outcome [18,19]. Preclinical studies are examining the role of neuronal ion transport channels, mitochondrial function, oxidative stress, and inflammation to develop novel approaches to ameliorate CIPN [18,19,20,21]. Oxidative stress appears to be central in mediating the therapeutic effect of chemotherapy in cancer. In contrast, oxidative stress is central to axonal injury. Therefore, it is likely that chemotherapeutic agents induce ER stress, which may contribute to oxidative stress and axonal injury leading to neuropathic symptoms.

## 3. CIPN Impairs Quality of Life

### 3.1. Sensory Changes

CIPN commonly affects 19% to 80% of individuals presenting with pain [22], sensory loss [23], paresthesia [24], dysaesthesia [25], numbness [24], poor dexterity [26], and discrimination of thermal information [9]. Symptoms often begin in the fingers and toes and spread proximally [27]. The onset and course of symptoms are variable, as it starts within weeks or months of therapy initiation and, in most cases, is partially reversible or possibly permanent [28]. The degree and severity of neuronal damage and its relative clinical symptoms depend on several risk factors, including the chemotherapeutic agent and associated drug combination, the dose and duration of therapy, in addition to individual predisposing factors that include age and preexisting neuropathic conditions, e.g., diabetes [29]. Symptoms can sometimes persist after discontinuation of chemotherapy. Thus, symptom presentation may be heterogeneous, but it impairs the quality of life. With an increasing number of cancer survivors, prevention of this disabling condition requires urgent attention.

### 3.2. Falls and Gait Changes

The accumulating neurotoxic effect of chemotherapeutics impacts motor skills and proprioception [30]. This is reflected in muscle weakness, balance disturbances, and loss of deep tendon reflexes [31,32]. Diminished sensation and loss of neuromuscular control of the lower extremities may escalate to interfere with balance and gait eventually [33]. Falls and unsteady gait are frequently reported by cancer patients with CIPN which could start even after one cycle of chemotherapy [34] and persist many years after discontinuation of treatment [35]. Falls are a considerable health concern, the leading cause of injury and death in older adults, including persons with cancer [36]. A study by Marshall et al. recently sought to identify the impact of CIPN on spatiotemporal gait parameters and fall risk [37]. The study concluded that cancer patients with CIPN may display slower gait velocities, shorter step lengths, and increased fall risk than healthy age-matched controls [37]. The underlying mechanism is not yet fully explored, and cerebellar affection might play a role evidenced by the loss of Purkinje cells in the cerebellum [38]. Interestingly, Purkinje cell damage and hyperalgesia have been demonstrated in transgenic mice with sickle cell disease (SCD), featuring neuropathic pain with several features reminiscent of CIPN [39,40]. Sickle mice showed altered stance instability and dynamic gait parameters similar to those found in chemotherapy-induced neuropathy. These alterations in gait correlated with hyperalgesia and were associated with cerebellar Purkinje cell apoptosis [41]. It is conceivable that motor dysfunction may be due to chemotherapy-induced damage to Purkinje cells and/or pain. Similar preclinical studies are needed to understand the neuronal pathobiology underlying CIPN involving the centrally mediated mechanisms.

## 4. Molecular Mechanisms Underlying CIPN

The known associated primary molecular mechanisms involve a decline in translation and transcription output [42], and axonal transport [43], alteration in ionic homeostasis sodium ion (Na^+^), calcium ion (Ca^2+^), and potassium ion (K^+^) [44], microtubular alterations [45], ER stress induction [46], mitochondrial function impairment [47], and disturbed oxidative homeostasis [9,48]. However, mechanism-based targeting of CIPN requires investigation to develop targeted therapies. Most of the mechanisms converge at the central role of reactive oxygen species (ROS), but antioxidants are not effective. Therefore, processes such as ER stress that contribute to ROS production require a deeper understanding of CIPN.

## 5. Endoplasmic Reticulum Stress as a Contributor to CIPN

### 5.1. ER Structure and Function

The structure and function of axonal ER are unique [49]. A separate characterization of axonal ER was first done in 1976 by Tsukita and Ishikawa [50]. The tapered, smooth endoplasmic reticulum (SER)/tubular ER of axons constitutes the central part of ER and functions in synthesizing lipids, regulating Ca^2+^ homeostasis [51,52], and controlling trafficking of biomolecules and ions [53,54]. On the other hand, the rough endoplasmic reticulum (RER) manages protein folding and export [55]. The mutations in ER regulatory genes are associated with several nervous system-associated pathologies, including Hereditary spastic paraplegia [56], Warburg Micro Syndrome [57], Charcot–Marie–Tooth disease (CMTD) [58], Parkinson’s disease (PD) [59], and Alzheimer’s disease (AD) [60].

### 5.2. Factors Regulating Oxidative Homeostasis in Axons under ER Stress

Imbalance in the oxidative state of axons underlies several nervous system-associated disorders [61]; we describe the key factors here:

#### 5.2.1. Misfolded Protein Response

Axonal ER under stress leads to misfolded proteins synthesized in the ribosomes, which cannot exit ER leading to the aggregation of misfolded proteins (MP) in its lumen, leading to the development of ER stress [62,63] (Figure 1). The sensing of accumulation of misfolded proteins involves three ER transmembrane sensors PERK (protein kinase RNA-activated-like ER kinase), IRE1α (inositol requiring enzyme 1α), and ATF6α (activating transcription factor 6α), and they start a cascade of molecular events inducing an unfolded protein response (UPR) to re-establish cellular homeostasis [64,65]. Prolonged ER stress activates c-jun N-terminal kinase (JNK) and p38 MAPK which can promote cellular apoptosis and necrosis [66]. Activated JNK inactivates Bcl-2 on the ER membrane as well as on the mitochondrial membrane by translocating to mitochondria, while promoting the pro-apoptotic members of Bcl-2 family leading to the release of cytochrome c from mitochondria and initiating the process of apoptosis [67]. Bcl-2 in ER is known to regulate Ca^2+^ efflux, whereas the pro-apoptotic members of Bcl-2 family, Bax and Bak promote Ca^2+^ release from ER leading to an influx of Ca^2+^ into the mitochondria resulting in mitochondrial swelling, disruption of mitochondrial membrane, release of pro-apototic factors into the cytosol and apoptosis. Additionally, ER stress also activates the pro-apoptotic caspase pathway leading to the activation of caspase-3 in a cytochrome-c independent manner. Thus, ER stress via Ca^2+^ release orchestrates mitochondrial involvement in the process of apoptosis (Figure 1). Most of the investigation in CIPN and neuropathies overall has mostly focused on the mitochondrial mechanisms, which are critical. However, to develop interventions based on apoptosis or oxidative stress targeting or co-targeting of ER stress is required for effective outcomes.

#### 5.2.2. Mitochondrial Calcium Signaling

The distinctive feature of axonal ER stress involves protein misfolding and trafficking [68], declined biosynthesis efficiency [62], altered ER dynamics, imbalance of oxidative homeostatic equilibrium [69], and impaired inter organelle Ca^2+^ signaling [63]. ER stress in neurons directly also governs the oxidative state of the mitochondria and vice versa. Both organelles are in constant cross-talk through mitochondria-associated membranes (MAM) [70]. At the molecular level, inter organelle cross-talk between mitochondria, lysosome, ER, and the nucleus is also initiated to restrict secondary damage due to ER stress [66,71,72]. Ca^2+^ is a crucial mediator in ROS signaling [73]. Interestingly, Ca^2+^ inhibits complex III of electron transport chain (ETC), a known point of ROS production [74]. In neuroblastoma cell lines, Ca^2+^ released due to ER stress leads to irregular calcium-mediated impulses in mitochondria, leading to increased production of ROS [56]. Thus, a synergism exists between mitochondria and ER in instigating oxidative stress via Ca^2+^ suggestive of co-targeting both the ER as well as mitochondria to inhibit axonal ROS.

#### 5.2.3. Glutathione Imbalance

In axonal ER stress, the limited intracellular antioxidant pool cannot neutralize ROS, causing an abrupt increase in ROS levels which may stimulate cell death signaling pathways [75,76]. Glutathione (GSH) provides axons’ predominant antioxidant defense [77]. The ratio of reduced (GSH) to oxidized (GSSG) within neuronal cells is critical, and an imbalance in their ratio with a relative increase in GSSG is a measure of oxidative stress [78]. Under conditions of prolonged oxidative stress, the level of free GSH is decreased, leading to increased ROS and irreversible cellular damage and death. The GSH/GSSG ratio imbalance is associated with neurodegenerative disorders, such as Parkinson’s and Alzheimer’s disease [79]. One of the main reasons for ROS generation is the result of upregulated proofreading of mismatched disulfide bonds of misfolded protein by the local enzymes protein disulfide isomerase (PDI) and endoplasmic reticulum oxidoreductase 1 (ERO1) [80]. Therefore, strategies to shift the balance toward upregulation of GSH are required, which can be accomplished with inhibition of ER stress.

#### 5.2.4. Autophagy

An alternative protective pathway is the activation of autophagy by UPR due to ER stress [81]. Autophagy is a cellular phenomenon by which defective proteins and organelles are degraded and recycled through intermediary autophagosomes associated with hydrolytic enzyme-filled lysosomes [82]. Autophagosomes can digest and break down malfunctioned mitochondria, a significant producer of ROS [83]. This process can reinstate the normal oxidative homeostasis and prevent the subsequent activation of cell death pathways [84]. However, prolonged ER stress-induced autophagy signaling becomes a destructive mechanism and leads to neuropathy [85].

#### 5.2.5. Bioenergetic Imbalance

Furthermore, ER stress in axons remains central to the bioenergetic imbalance [66,71,72]. Enzymes of the mitochondrial matrix are Ca^2+^ sensitive, and the influx of excessive calcium ions from the ER can lead to modulated enzymatic activity [86]. This may directly lead to altered mitochondrial function, especially adenosine triphosphate (ATP) production efficiency in the cells [71]. In case of ER protein misfolding, there is a high demand for ATP from oxidative phosphorylation in mitochondria [87]. Therefore, on the one hand, the demand for ATP is increased, but its availability is impaired due to altered mitochondrial function, leading to the inability to meet ATP requirement for normal axonal maintenance.

## 6. Preclinical Evidence of ER Stress in CIPN

Vincristine injected in rats caused ER stress by an unfolded protein response (UPR), upregulation of PERK, IRE1, ATF-6, and chaperone glucose-regulating protein 78 (GRP78) that belong to Heat Shock Protein 70 family (Hsp70), followed by an increase in apoptotic proteins BCL-2 associated X (Bax), caspase 3 and a decrease in anti-apoptotic molecule B-cell lymphoma 2 (Bcl-2) [88]. In several preclinical studies in laboratory animals and cell lines, the modulation of chaperons Hsp29, Hsp90, and Hsp70 has shown significant therapeutic potential in treating CIPN [89,90,91]. Bortezomib-induced ER stress is known to cause peripheral neuropathy [15]. The chain of molecular events starting with UPR is followed by the segregation of chaperone GRP78/BiP (binding immunoglobulin protein) and PERK, thereby activating PERK, followed by phosphorylation of eIF2α (eukaryotic translation initiation factor 2α), which inhibits translation and protein folding (Figure 1) [15]. GRP78/BiP also dissociates from IRE1α, which mediates mRNA splicing reaction for the transcription factor X-box binding protein 1 (XBP1). In turn, this participates in the increased production of ER chaperones and CAAT/enhancer-binding protein homologous protein (CHOP), which contributes to ER-dependent degradative processes, including cell death if ER stress is severe and prolonged. ATF6 translocates to the Golgi apparatus, producing another transcription factor ATF6N attenuating proinflammatory pathways based on nuclear factor kappa light chain enhancer of activated B cells (NF-κB), p38 Mitogen-activated protein kinase (p38MAPK), and c-Jun N-terminal kinase (JNK) signaling [15]. ER stress molecular pathways are involved in taxane-induced apoptotic cell death [92,93]. The mechanism of paclitaxel-induced ER stress includes significant upregulation of phospho-eIF2a, CHOP, and apoptotic proteins caspase 4 and caspase 3 [94].

## 7. ER Stress in Disorders of the Nervous System

Axonal ER stress is noticed in several neurological disorders [95]. ER stress is central to the development of several neuropathic disorders, including diabetes-induced neuropathy [96,97,98], human immunodeficiency virus 1 (HIV-1)-associated inflammation, and antiretroviral treatment (ART), leading to ER stress and activation of UPR signaling in astrocytes [99,100], SCD-induced neuropathy [101,102], cerebrovascular dysfunction [103,104], and CMTD [105].

In the spinal cord microglial cells of mice with SCD, ER stress and ROS generation were mediated by the activation of Toll-like receptor (TLR4) by cell-free heme released due to ongoing hemolysis [101]. Further, either inhibition of TLR4 or ER stress led to a significant decrease in mechanical and thermal hyperalgesia. Hence, this study clearly shows that increased ER stress contributes to mechanical and thermal hyperalgesia in SCD mice, which resembles neuropathic symptoms of CIPN [101]. These findings are also in line with our previous observations that increased spinal ROS, substance P, microglial activation, p-STAT3 (phosphorylated signal transducer and activator of transcription 3), and stress-induced p38MAPK in association with hyperexcitability of spinal neurons in sickle mice [102]. We also observed that mast cell activation induced ER stress in the vascular endothelial cells leading to increased P-selectin expression and increased blood-brain barrier (BBB) permeability [103]. Notably, salubrinal, which reduces ER stress, inhibited BBB permeability in sickle mice, suggesting that ER stress may contribute to the central nervous system (CNS) pathology [103]. Many features of these nervous system disorders are common with CIPN, involving the CNS, in addition to the peripheral nervous system. Therefore, it is likely that ER stress may modulate CNS function such as the Purkinje cell apoptosis discussed above, BBB permeability, or pain inhibitory pathways, which may, in turn, contribute to CIPN.

## 8. Current Therapy and Phytochemicals Targeting ER Stress for CIPN

### 8.1. Duloxetine

Duloxetine, a serotonin-norepinephrine reuptake inhibitor, has been used to manage depression and anxiety [106]. It acts by inhibiting the reuptake of serotonin and norepinephrine, restoring their activity at the synapse. Additionally, serotonin and norepinephrine are involved in the descending inhibitory nociceptive pathway and thereby inhibiting pain via a top-down mechanism in the brain. Duloxetine has been recommended for the treatment of established CIPN, but the benefit remains limited [18,19]. It is therefore critical to examine the centrally mediated mechanisms of CIPN. Chemotherapeutic drugs that can cross the blood brain barrier may have a greater centrally mediated effect via top-down mechanisms and/or via Purkinje cells discussed earlier affecting gait.

### 8.2. Other Drugs under Study for CIPN

In this review, we have explored the role of ER stress as a possible underlying mechanism and thus a therapeutic target for CIPN. Traditional medicinal herbs and herbal formulas have shown positive outcomes in preclinical studies, as summarized in Table 1. It is worth noting that multiple drugs have been studied for their effect against CIPN with controversial results. Nerve-protective agents as acetyl-L-carnitine and glutamine have been used in several clinical trials with various chemotherapeutic agents, including taxane [107], docetaxel, and paclitaxel [108,109,110], with little or no effects. Other clinical trials targeted oxidative stress using antioxidants like vitamin E [111], A-Lipoic acid [112], and vitamin B [113] with no significant benefits on pain or functional outcomes. Preclinical studies using anti-inflammatory, e.g., minocycline [114] that inhibit the activation of monocytes and metformin that activates AMPK (AMP-activated protein kinase) [115] and ion channel targeted agents [116,117], have promising results in animals that need to be verified for efficacy and safety in humans. The development of novel approaches is needed to target complex mechanisms such as ER stress which contribute to oxidative stress but are not responsive to antioxidants. Considering the toxicity of chemotherapeutic agents, developing approaches without the adverse effects of pharmacologics are needed to treat CIPN by targeting ER stress.

### 8.3. Phytochemicals to Target ER Stress in CIPN

Phytochemicals are being significantly used in preclinical studies to treat CIPN [127,128], but literature on the use of phytochemicals to target ER stress for CIPN remains scarce. Paclitaxel induced ER stress by increasing PERK, IRE1, ATF6, GRP78, and CHOP mRNA transcript levels in the sciatic nerve of Sprague-Dawley rats [129]. Hesperidin, a bioflavonoid with antioxidant and anti-inflammatory effects found ubiquitously in citrus fruits [130], ameliorated paclitaxel-induced ER stress and associated neuropathic pain [129]. Aucubin, an iridoid glycoside, and the active constituent of a traditional Chinese medicinal herb extract of *Plantaginis semen* [131] prophylactically attenuated paclitaxel-induced mechanical allodynia in mice [132]. Co-treatment of mice with paclitaxel and Aucubin for 14 days significantly inhibited the expression of CHOP in the sciatic nerve [132]. This in vivo effect of Aucubin was validated in a Schwann cell line, LY-PPB6, in vitro. In a dose-dependent manner, Aucubin significantly inhibited paclitaxel-induced CHOP expression in LY-PPB6 cells. Together, these in vivo and in vitro data suggest that Aucubin ameliorates mechanical allodynia via inhibition of ER stress [132].  Further examination of these phytochemicals is required for clinical translation.

## 9. Phytochemicals Targeting ER Stress in Nervous System Disorders

Since studies on CIPN are limited, we describe several related nervous system-associated pathologies that have examined the effect of phytochemicals/nutraceuticals on ER stress-mediated targeting (Table 2).

### 9.1. Resveratrol (3,5,4′-Trihydroxy-Trans-Stilbene)

A known plant-based antioxidant found in the skin of berries, red grapes, and fruits [133]. When combined with vitamin D, the resveratrol leads to downregulation of ER stress markers GRP78, p-eIF2α, and CHOP in SH-SY5Y cells in an in vitro microenvironment of AD [134]. In PD model organism, resveratrol inhibited rotenone-induced ER stress markers CHOP and GRP78 levels followed by the downregulation of caspase-3 activity in the brain of rotenone-exposed rats [135]. The decrease in ER regulatory mechanisms was accompanied by a reduction of caspase-3 of the apoptotic pathway. The protective property of resveratrol against PD appears to involve controlling apoptosis and inflammation via suppressing ER stress in the brain [135]. In Batten disease (neuronal ceroid lipofuscinosis), ER stress is one of the prominent causes of oxidative dysfunction and cell death [136]. The resveratrol treatment in vitro increased the superoxide dismutase (SOD) activity and lowered the GRP78 levels. Finally, downregulation in the expression level of apoptosis markers such as apoptosis inducible factor (AIF) and cytochrome c was reported [136]. In postoperative cognitive dysfunction (PCOD), resveratrol activates Sirt (silent mating type information regulation 2 homolog), a pivotal player in protecting mice from cognitive impairment after surgery [137]. Inhibiting various ER stress pathways by resveratrol leads to downregulation of GRP78, IRE1, XBP1, PERK, and CHOP, which strongly indicates that counteracting ER stress is a possible target in preventing POCD [138]. The significance of ER stress in nociception has been reported for several neuropathies [85,97]. The possible protection of resveratrol against vasculitis-induced neuropathy has been investigated; it suppressed ER stress-induced NF-κB-mediated neuroinflammation [139]. Subarachnoid hemorrhage (SAH) can lead to stress induction in ER [140]. The therapeutic value of resveratrol in SAH was investigated in rats [141,142]. Not only did resveratrol elevate the survival rate, but it also lowered brain edema and apoptosis in neurons [141,142]. Besides strengthening the intrinsic antioxidant properties of the brain, resveratrol downregulated GRP78 and CHOP expression [141]. In cerebral ischemia, resveratrol reduced the infarct area due to activation of the estrogen and N-methyl-D-aspartate (NMDA) receptors [143]. There was also evidence that the increase in ER chaperones (GRP78 and GRP94) due to resveratrol pretreatment may have decreased the load of misfolded proteins in ischemic tissue [144]. Hence, resveratrol treatment might prevent neural injury under ischemia-reperfusion via increasing endoplasmic reticulum-associated protein degradation (ERAD). 

### 9.2. Berberine (9,10-Dimethoxy-5,6-Dihydro-2H-7λ5-[1,3] Dioxolo[4,5-g]Isoquinolino[3,2-a]Isoquinolin-7-Ylium)

It is extracted from the plants of Coptis sp. and Berberis sp. [145]. The inhibition of ER stress with berberine in APP/PS1 transgenic mice has been found to ameliorate the pathology of AD [146,147]. The molecular mechanism associated with the treatment involves inhibiting PERK/eIF2α signaling-mediated Beta-Secretase 1 (BACE1) translation, thus reducing Aβ production and resultant neuronal apoptosis [147]. In another study, berberine improves diabetic encephalopathy through the SIRT1/ER stress pathway in db/db mice and inhibited the expression of ER stress-associated proteins (PERK, IRE-1α, eIF-2α, PDI, and CHOP) in the hippocampus of db/db mice [146,148].

### 9.3. Curcumin(1E,6E)-1,7-bis (4-Hydroxy-3-Methoxyphenyl)-1,6-Heptadiene-3,5-Dione)

A known neuroprotective agent and has been recently associated with inhibition of ER stress regulatory factors in several nervous system-associated disorders [149,150]. The CMTD [CMT1 (CMT1A and CMT1B)] is associated with unregulated activation of ER stress-mediated UPR leading to cellular damage [151]. In the course of treatment, curcumin is found to impair calcium-dependent chaperones (calnexin, calreticulin), thus altering the ER Ca^2+^ levels and subsequently decreasing the UPR/activation of the ERAD pathway [152]. In diabetes-related cognitive dysfunction, combination of exercise and curcumin improved the escape latency and memory retention by negatively altering levels of ER stress markers BiP and CHOP [150]. In another study, curcumin inhibits ER stress-associated activation of TXNIP/NLRP3 (Thioredoxin-Interacting Protein/Nod-like Receptor Protein 3) inflammasome and attenuates glutamate-induced neurotoxicity in the hippocampus [153]. Curcumin also suppresses ER stress induced by cerebral ischemia-reperfusion injury in rats [154]. Administering curcumin in the neuropathic mice model Tr-J downregulates expression of UPR markers indicative of ER stress sensors alleviation of ER stress in sciatic nerves of mice, and hence improving the symptoms of neuropathy [155].

### 9.4. Epigallocatechin (Epigallocatechin-3-Gallate, EGCG)

EGCG is a green tea catechin [156]. The therapeutic efficacy of EGCG in ER stress-related disorders shows promise. EGCG reduced ER stress in transient focal cerebral ischemia in a rat model and AD in an in vitro study [157,158]. It involves inhibiting glucose-regulated ER stress regulators caspase-12, CHOP, and GRP78 [157,158]. This process was subsequently followed by the inhibition of Transient Receptor Potential Cation Channel Subfamily C Member 6 (TRPC6) proteolysis, along with activation of the cAMP-response element binding protein (CREB) downstream of MAPK/ERK (extracellular signal-regulated kinase) pathway [158]. Interestingly, EGCG also has been found to act antagonistically to the two prominent ER stress booster drugs, tunicamycin and thapsigargin by promoting autophagy-dependent survival through modulation of mTOR-AMPK pathways upon ER stress [159]. The results of ER stress regulation by EGCG in transient focal cerebral ischemia are also supported with an in vivo study in rats [158]. In AD, EGCG reduces neurotoxicity caused by β-amyloid, and induced a decline of ER stress-mediated apoptosis, as documented in [157].

### 9.5. Ginsenoside Rb1 (C54H92O23)

One of the active ingredients in traditional herb ginseng [160]. The ER stress relieving mechanism involves PERK and CHOP downregulation in high glucose-treated hippocampal neurons [161]. The ginsenoside Rb1 inhibition of ER stress is similar to 4-PBA (4-phenylbutyric acid) and antioxidant N-acetyl cysteine (NAC) [162]. Moreover, ginsenoside Rb1 modulates the elevation of intracellular ROS and the disruption of mΨ induced by high glucose [161,163]. In another study, ginsenoside Rb1 modulates the cellular antioxidant defense and shows a partial cytoprotective role in dopaminergic cell culture systems through ER-dependent Heme oxygenase-1 (HO-1) induction via the Gβ1/PI3K/Akt-Nrf2 axis of the signaling pathway [164]. It has also been found to modulate formaldehyde-related cytotoxicity to astrocytes of CNS [162]. The glucose-regulated ER stress modulators GRP78 and CHOP translatory expression were increased, and pro-caspase-12 was decreased in PC12 pheochromocytoma cells after exposure to formaldehyde upon treatment with ginsenoside Rb1 [162].

## 10. Conclusions and Future Perspectives

ER stress contributes to neural injury and may underlie the pathobiology of CIPN, thus providing a mechanism-based target for developing novel therapeutic strategies. Integrative strategies using phytochemicals to inhibit ER stress may modulate the oxidative imbalance in the nervous system, thus providing protection from ROS-induced injury. Since traditional antioxidants have not been successful at ameliorating CIPN, inhibition of ER stress provides a novel target for preventing and/or treating CIPN. The use of integrative approaches to use phytochemicals for targeting ER stress may be devoid of the cytotoxic effects and may therefore ameliorate CIPN without adverse effects.

## Figures and Tables

**Figure 1 antioxidants-11-00265-f001:**
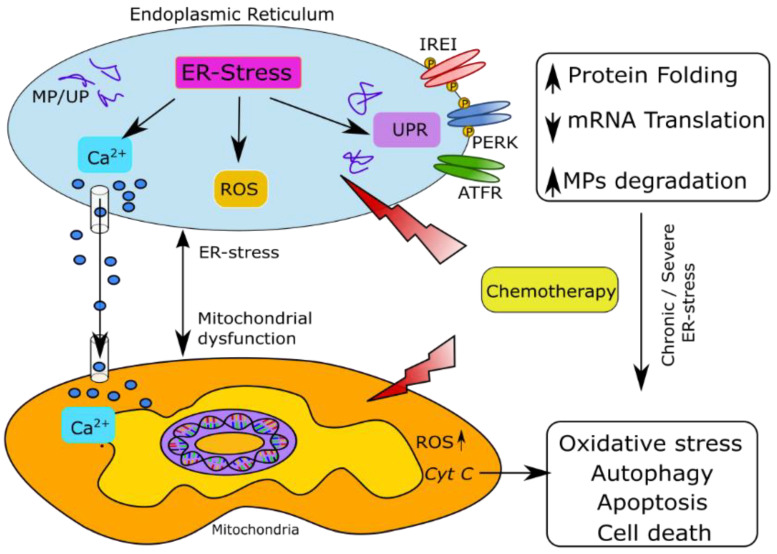
Endoplasmic reticulum stress contributes to chemotherapy-induced peripheral neuropathy. Normal functions of ER include protein synthesis, maturation, and the folding of proteins and transport. Disturbance of these functions by chemotherapy may induce ER stress to kill tumor cells, resulting in the accumulation of misfolded proteins. ER stress leads to the activation of unfolded protein response (UPR) via UPR sensors: ATF6α, IRE1α, and PERK. This further activates signaling pathways as adaptive response to reduce the protein misfolding. The UPR enhances the ER capacity for protein folding and modification, attenuates global mRNA translation, and disposes misfolded proteins by ER-associated protein degradation (ERAD) and autophagy. If ER stress persists, the UPR triggers cell death. ROS are generated as a byproduct of oxidative protein folding catalyzed by ER oxidoreductases, including protein disulfide isomerases (PDI). Accumulated misfolded proteins increase the protein-folding load and consume the antioxidant reserve in the cell. Mitochondrial ROS are generated by calcium-enhanced ER release. Elevated Ca^2+^ levels in the mitochondria cause the release of cytochrome c and subsequently trigger death signals via caspase activation. Together, these processes may cause neural injury, leading to neuropathic symptoms of chemotherapy. Abbreviations: ATF6α: activating transcription factor 6α, Ca^2+^: calcium ion, ER: endoplasmic reticulum. ERAD: ER-associated protein degradation IRE1α: inositol requiring enzyme 1α, MPs: misfolded protein, PDI: protein disulfide isomerases PERK: protein kinase RNA-activated-like ER kinase ROS: Reactive Oxygen Species, UP: unfolded protein, UPR: unfolded protein response.

**Table 1 antioxidants-11-00265-t001:** Investigational phytochemicals to treat CIPN.

Medicinal Herbs and Formulas	Chemotherapeutic Agent	Preclinical Outcomes	References
Acorus calamus	Vincristine	Attenuated vincristine-induced thermal and mechanical hyperalgesia, biochemical and histopathological changes via its anti-oxidative, anti-inflammatory, neuroprotective, and calcium inhibitory actions.	[118]
Cannabinoids	Vincristine	Inhibited vincristine-induced mechanical allodynia through activation of CB1 and CB2 receptors at the spinal cord level.	[119]
Ocimum sanctum L	Vincristine	Attenuated vincristine-induced neuropathic pain and decreased oxidative stress and calcium levels.	[120]
Salvia officinalis (Sage)	Cisplatin	Anti-nociceptive effect on cisplatin-induced hyperalgesia in mice comparable to morphine injections.	[121]
Walnut	Cisplatin	Walnut consumption improved memory and motor abilities in cisplatin-treated rats.	[122]
Jesengsingi-Hwan (Goshajinkigan in Japanese)	Oxaliplatin	Relieved the oxaliplatin-induced cold hyperalgesia and mechanical allodynia without affecting the anti-tumor activity of oxaliplatin.	[123]
	Paclitaxel	Markedly prevented paclitaxel-induced mechanical allodynia.	[124]
Gyejigachulbu-tang	Oxaliplatin	Relieved oxaliplatin-induced cold and mechanical hyperalgesia possibly by suppression of spinal glial activation.	[125]
Jakyakgamcho-Tang	Paclitaxel	Significantly improved the paclitaxel-induced allodynia and hyperalgesia.	[126]

**Table 2 antioxidants-11-00265-t002:** Phytochemicals targeting endoplasmic reticulum stress in nervous system-associated disorders.

Phytochemical	Major Source	ER Stress Regulators	Disorder	References
Hesperidin	*Citrus aurantium*	PERK↓ IRE1↓ ATF6↓ GRP78↓ CHOP↓	Chemotherapy Induced peripheral neuropathy	[129]
Aucubin	*Plantaginis semen*	CHOP↓	Chemotherapy Induced peripheral neuropathy	[132]
Resveratrol	*Polygonum cuspidatum*	GRP78↓ CHOP↓ XBP1↓ p-eIF2α↓ PERK↓	Alzheimer’s disease	[134]
		GRP78↓ CHOP↓	Parkinson’s disease	[135]
		GRP78↓	Batten disease	[136]
		XBP1↓ PERK↓ IRE1↓	Post-Operative Cognitive Dysfunction	[138]
		IRE1↓	Neuropathy	[139]
		GRP78↓	Cerebral ischemia	[143]
Berberine	*Berberis vulgaris*, *Berberis aristata*	PERK↓, eIF2α↓	Alzheimer’s disease	[147]
		PERK↓, IRE1α↓, eIF-2α↓, PDI↓, CHOP↓	Diabetic Encephalopathy	[148]
Curcumin	*Curcuma longa*	BiP↓ CHOP↓	Diabetes-related cognitive dysfunction	[150]
		BiP↓ CHOP↓	Charcot-Marie-Tooth disease	[155]
Epigallocatechin-3-gallate	*Camellia sinensis*	GRP78↓ CHOP↓	Alzheimer’s disease	[157]
		GRP78↓ CHOP↓	Cerebral ischemia	[158]
Ginsenoside Rb1	*Panax ginseng*	PERK↓, CHOP↓	high glucose-treated hippocampal neurons	[161]
		PERK↓, CHOP↓ GSK3β↓	Formaldehyde induced neurotoxicity	[162]

Abbreviations: BiP: binding immunoglobulin protein, CHOP: C/EBP homologous protein, GRP78: glucose-regulating protein 78, GSK3β: glycogen synthase kinase-3β, IRE-1α: inositol requiring enzyme 1α, PDI: protein disulphide isomerase, p-eIF2α: eukaryotic translation initiation factor 2A, PERK: protein kinase RNA-activated-like ER kinase, and XBP1: X-box binding protein 1.

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
