# Peer review of "Endoplasmic Reticulum Stress in Chemotherapy-Induced Peripheral Neuropathy: Emerging Role of Phytochemicals"

_antioxidants, 2022, doi:10.3390/antiox11020265_

Round 1
Reviewer 1 Report
This review focuses on the impact of ER stress in chemotherapy-induced peripheral neuropathy (CIPN).
Authors aim to bridge the crucial gap regarding ER contribution in CIPN.
This study is important in the field. However, in this version, the manuscript appears disorganized. The information is not exhaustive. Almost always, the literature is only listed without a critical discussion. In addition, the manuscript needs a general editing.
In this version, the manuscript appears little appropriate for publication in Antioxidants. (SI: Natural or Synthetic Antioxidants for Neurodegenerative Diseases and Brain Health)
Author Response
Reviewer 1
This review focuses on the impact of ER stress in chemotherapy-induced peripheral neuropathy (CIPN). Authors aim to bridge the crucial gap regarding ER contribution in CIPN. This study is important in the field. However, in this version, the manuscript appears disorganized. The information is not exhaustive. Almost always, the literature is only listed without a critical discussion. In addition, the manuscript needs a general editing.
Author response: Our apologies for the oversight on a systematic organization. We appreciate the Reviewer’s suggestions and reorganized the entire manuscript by first providing insights on the problem of CIPN, its current therapeutic strategies, and the possible contribution of ER stress, followed by other neuropathic conditions with similarity to CIPN and finally the potential of phytochemicals for targeting ER stress and CIPN.
In this version, the manuscript appears little appropriate for publication in Antioxidants. (SI: Natural or Synthetic Antioxidants for Neurodegenerative Diseases and Brain Health).
Author response: We posit that oxidative stress is extremely critical to CIPN and ironically brain mediated mechanisms in CIPN remain underexamined. In the manuscript throughout, we have highlighted the paucity of CNS-based mechanisms. Our hope is that the publication of this manuscript will lead to extending the focus on CNS mediated mechanisms including Purkinje cell damage and top-down inhibitory nociceptive mechanisms and targeting of ER stress as a co-contributor to oxidative stress. It also highlights the significance of advancing antioxidant-based therapies to target ER stress in addition to mitochondria-based therapies.
Reviewer 2 Report
Interesting review on a underestimated severe clinical problem (CIPN).
There are a few comments which might further improve the review
The function of Dulexetine should be mentioned in the introduction as well as the mechanism of action of the cytostatic drugs which are mentioned.
The authors should mention as to whether other ER or non ER chaperones (aprt from Grp78) are involved in CIPN.
Chapter 6.1. can the authors specify the cell death mechanisms more detailed.
Chapter 9 should either be extended or deleted. The ER stress should be discussed more specifically for different tumor types.
The function of GSH in the Redox system should be mentioned briefly.
Apart from Phytochemicals other drugs which interfere with CIPN should be included.
I would suggest to change the title and include the role of phytotherapies in adressing CIPN. However, a chapter needs to be included about traditional drugs in the treatment of CIPN.
Author Response
Reviewer 2
Comment 1: “The function of Duloxetine should be mentioned in the introduction as well as the mechanism of action of the cytostatic drugs which are mentioned”
Author response: Thank you for your suggestion, and we have added more details in the introduction to reflect the mechanism of action of cytostatic drugs (P.2 line 69). We have included duloxetine in the Introduction and also elaborated on its action in section 8.1, page10.
Comment 2: “The authors should mention as to whether other ER or non ER chaperones (aprt from Grp78) are involved in CIPN”
Author response: This is a very nice suggestion. In the revised manuscript we present a description of modulation of significant chaperons Hsp27, Hsp70, and Hsp90. Indicated in red and incorporated within section 6, page 7.
Comment 3: “Chapter 6.1. can the authors specify the cell death mechanisms in more detail?”
Author response: A detailed description has been incorporated in the main text of the manuscript along with references in section 5.2-[i], page 5.
Comment 4: “Chapter 9 should either be extended or deleted. The ER stress should be discussed more specifically for different tumor types.
Author response: This section has been deleted as suggested, in the revised manuscript.
Comment 5: “The function of GSH in the Redox system should be mentioned briefly.”
Author response: Greater insight into GSH and its role in the redox system, has been incorporated in the manuscript along with references in section 5.2-[iii] page 6 of the manuscript.
Comment 6: “Apart from Phytochemicals other drugs which interfere with CIPN should be included”
Author response: Section 8.2 page 10 has been added to include other drugs that interfere with CIPN from clinical and preclinical studies.
Comment 7: “I would suggest to change the title and include the role of phytotherapies in adressing CIPN. However, a chapter needs to be included about traditional drugs in the treatment of CIPN.”
Author response: The title has been changed to “Endoplasmic reticulum stress in chemotherapy-induced peripheral neuropathy: Emerging role of phytochemicals.” We agree that it is more reflective of the contents of the review.
New Table1, page 11 has been added in section 8.2 to summarize the medicinal herbals and herbal formulas that have been applied in preclinical experiments to treat CIPN with different chemotherapeutic agents and the most significant outcome. Table 1 in the original manuscript is now Table 2 in the revised manuscript.
Round 2
Reviewer 1 Report
Report after revision
This review provides a focus on the impact of ER stress in chemotherapy-induced peripheral neuropathy (CIPN). This is a very important issue in the field since a combination between therapeutic approaches converging to target both the ER and mitochondrial network could improve treatments for CIPN.
I think that this revised version of the manuscript is now recommended for publication in Antioxidants. (SI: Natural or Synthetic Antioxidants for Neurodegenerative Diseases and Brain Health)